# Bioequivalence Analysis of Terazosin Hydrochloride Tablets Based on Parallel Artificial Membrane Permeability Analysis

**DOI:** 10.3390/ph17081024

**Published:** 2024-08-05

**Authors:** Jianzhao Niu, Hanhan Huang, Ming Ji, Wenjing Zhang, Yin Huang, Lingyun Ma, Baolian Wang, Qian Liu

**Affiliations:** 1NMPA Key Laboratory for Quality Research and Evaluation of Chemical Drugs, National Institutes for Food and Drug Control, Beijing 100050, China; njz@nifdc.org.cn (J.N.);; 2School of Pharmacy, China Pharmaceutical University, Nanjing 210009, China; 3Institute of Materia Medica, Chinese Academy of Medical Sciences & Peking Union Medical College, Beijing 100050, China

**Keywords:** terazosin hydrochloride tablets, bioequivalence, flux, permeability, dissolution profiles

## Abstract

Parallel artificial membrane permeability analysis (PAMPA) is used to determine the permeability of compounds through concentrated negatively charged phospholipid bilayer barriers. We employed MacroFlux (a scaled-up version of PAMPA) to test the permeation rate of terazosin hydrochloride (TH) tablets and predict in vivo bioequivalence. The dissolution profiles and permeability of one reference formulation, and seven generic TH tablets, were compared. The dissolution profiles of these generic tablets were equivalent to that of the reference drug in four different media. However, the flux and the total permeated amount of some generic TH tablets were below the lower limit of the confidence interval of the original acceptance range in MacroFlux, which implied risk in the bioequivalence test in vivo. We further evaluated potential factors responsible for this discrepancy by µFlux, including active pharmaceutical ingredient (API) permeability and excipient prescriptions. The analysis showed that different properties of API were a main factor leading to biological inequivalence in the MacroFlux assay, while excipient prescriptions did not have an impact on bioequivalence risk. These data indicated that the flux assay may be a helpful as an auxiliary method for predicting bioequivalence of generic drugs and analyze the factors responsible for bioequivalence risk.

## 1. Introduction

Generic drugs refer to products that contain the active pharmaceutical ingredient (API) and are prepared using the same active composition, dosage form, specification, indication, administration route, usage, and dosage as the original research drug [1]. China is a large manufacturer of generic drugs, accounting for 95% of generic drugs [2]. Generic drugs play an important role in the basic medical supply and meeting the public demand for clinical medications [3]. Therefore, the consistent evaluation of generic drug quality and efficacy is of great significance for improving the quality of generic drugs and ensuring drug safety [4,5]. The aim of this approach is to make the quality and efficacy of generic drugs consistent with those of the original drug [6].

Terazosin hydrochloride (TH) is a selective α1 receptor blocker, which can relax blood vessels and prostate smooth muscle and has a good safety profile. It is widely used in the treatment of mild-to-moderate hypertension and benign prostatic hyperplasia [7,8,9,10]. It belongs to the first class of Biopharmaceutical Classification System (BCS), characterized by high solubility and permeability. This drug was successfully developed by Abbott Laboratories (Green Oaks, IL, USA) and was first launched in Germany in 1985 with the trade name Hytrin^®^. The commonly used dosage forms include tablets and capsules.

Clinically, the most common form of administration is oral formulation, which is mainly absorbed in the intestine. The drug is absorbed by the intestinal tract into the blood circulation and subsequently distributed to various organs and tissues to reach the lowest effective concentration and exert its therapeutic effect. The absorption of oral solid preparations in the intestine is mainly related to the dissolution behavior of preparations and the permeability of drugs [11]. Drugs enter the blood mainly through passive transport, active transport, bypass transport, etc. Of note, approximately 95% of drugs permeate into the blood through passive transport across small intestinal epithelial cells [12]. There are many methods for the determination of drug permeability, including in vivo and in vitro tests [13,14], such as a mass balance test, absolute bioavailability test, and human intestinal perfusion test, as well as in vivo or in situ intestinal perfusion test suitable for animal models.

The in vitro tests include an in vitro tissue transmembrane test, a monolayer epithelial cell test, and artificial membrane models. Parallel artificial membrane permeability analysis (PAMPA) is a method that can be used to determine the permeability of compounds through concentrated negatively charged phospholipid bilayer barriers [15]. It is characterized by high throughput, low cost, and convenient detection methods, and it is widely used in drug in vitro permeation tests. Models such as the gastrointestinal absorption model [16], the blood–brain barrier-absorption model [17], the corneal absorption model [18], and the skin absorption model [19] were established [20]. Moreover, it has been shown that the gastrointestinal permeability results measured by PAMPA were positively correlated with those measured by the human small intestinal perfusion method [21].

In this study, a scaled-up version of PAMPA was used to determine the permeability of TH and the flux of TH tablets. This approach could potentially combine dissolution and absorption testing [22]. The MacroFlux apparatus was used to determine the flux and total permeated amount of a TH reference formulation and generic formulations under fasted state simulated intestinal fluid (FaSSIF) and fed state simulated intestinal fluid (FeSSIF) condition, as well as to predict their bioequivalence. The formulations that did not meet the bioequivalence were analyzed by the µFlux apparatus. The in vitro permeation model of TH tablets was established, which is helpful for consistency evaluation and bioequivalence study.

## 2. Results

### 2.1. In Vitro Dissolution Profiles of TH Tablets

We initially compared the in vitro dissolution profiles of these tablets. According to the Statistical Guidelines for Bioequivalence Studies [23], the main pharmacokinetic parameters of generic and reference formulations should be between 80% and 125%, which can preliminarily predict the bioequivalence of generic and reference formulations. The dissolution profiles of each tablet in four media are shown in Figure 1. The results indicated that the dissolution of all samples was >85% at 15 min, without the need to calculate the f_2_. These findings suggested that the dissolution profiles of TH reference and generic formulations were consistent.

### 2.2. MacroFlux Assay of TH Tablets

As TH is classified as BCS Class I, the MacroFlux assay could be used to investigate the flux and total permeated amount under FaSSIF and FeSSIF conditions, and the confidence interval of the generic formulations was calculated to predict the risk level of the generic formulations in the bioequivalence test. The MacroFlux experiments lasted for 4 h. The flux and total permeated amount of TH tablets were calculated between 100–200 min and 100–240 min under FaSSIF and FeSSIF conditions, respectively (Table 1).

According to Table 1, the flux and total permeated amount of TH tablets from Enterprises A, C, E, and F were consistent with those of the reference formulation under FaSSIF and FeSSIF conditions. The 90% confidence interval of all bioequivalence parameters was within the range of 80–125%, indicating that the TH tablets were equivalent to the reference formulation and had a high probability of passing the bioequivalence test. The flux and total permeated amount of TH tablets provided by Enterprises B and G were lower than those of the reference in FaSSIF and FeSSIF conditions, and the lower limit of the confidence interval was not within the original acceptance range (80–125%). It was preliminarily predicted that the TH tablets produced by these two enterprises were more likely to fail the bioequivalence test (Table 2). The 90% confidence interval of the flux of TH tablets D in the FeSSIF medium was not within the range of 80–125%, while the flux and total permeated amount in the FaSSIF state were basically consistent with those of the reference. The data indicated a risk in the satiety bioequivalence test for TH tablets from Enterprise D. To investigate the reasons responsible for the high risk of TH tablets from Enterprises B, D, and G in the bioequivalence test, this experiment was conducted from the two aspects of API permeability and excipient prescriptions.

### 2.3. Permeability of API

Among TH tablets provided by eight enterprises, the API of TH tablets produced by Enterprise D was from Enterprise I, while TH tablets produced by other companies came from II. The permeation curves of TH API provided by Enterprises I and II under the conditions of pH 5.0, 6.5, and 7.4 are shown in Figure 2. The concentration of TH API in the acceptor chamber gradually increased, and under the conditions of the three pH values, the penetration of TH API provided by Enterprise II was faster than that provided by Enterprise I.

According to Equation (5), the permeability of TH API provided by the two enterprises was calculated under these three pH conditions (Table 3). There was no significant difference in the permeability of TH API produced by the two enterprises under the conditions of pH 6.5 and 7.4 (*p* > 0.05). Under the condition of pH 5.0, the permeability of TH API from Enterprise I was slower than that from Enterprise II, and the difference was significant (*p* < 0.05). These data indicated that different properties of API were one of the main factors leading to bioequivalence risk in the MacroFlux assay.

### 2.4. Permeability of the Powder of TH Tablets

Next, we evaluated the excipient prescriptions in this assay. The results of the powder of TH tablets with bioequivalence risk under FaSSIF and FeSSIF conditions are shown in Table 4.

The influence of the formulation process in the experiment was avoided by grinding the tablets into powder. Under the FaSSIF condition, the permeability of the reference formulation was slightly higher than that of the generic formulation. Under the FeSSIF condition, the permeability of the reference formulation was lower than that of the generic formulations. However, there was no statistically significant difference (*p* > 0.05). Therefore, the difference in excipient compositions did not cause bioequivalence risk.

### 2.5. Bioequivalence Studies of TH Tablets in Dogs

To further confirm whether the generic and reference formulations were bioequivalent, we performed an in vivo assay in beagle dogs. Blood concentration and related pharmacokinetic parameters of the generic formulation (Enterprise B) and the reference formulation were detected. The 90% confidence interval of the AUC or C_max_ was used to assess the bioequivalence of the generic drug (Enterprise B). The blood concentration-time curve of six beagle dogs following oral administration of TH generic formulation and reference formulation is shown in Figure 3.

The main pharmacokinetic parameters are shown in Table 5. The 90% confidence intervals for C_max_ and AUC of the generic formulation were 84.57–122.60% and 91.68–104.87% of the corresponding parameters of the reference formulation, respectively. This finding met the requirement that the 90% confidence interval must fall within the range of 80.00–125.00% for dogs with non-narrow therapeutic windows. Thus, the results obtained in vivo indicated that the reference formulation and the generic formulation were bioequivalent in beagle dogs.

## 3. Discussion

### 3.1. In Vitro Dissolution Experiment

The dissolution experiment is used to simulate the disintegration and dissolution of oral solid preparations in the GIT in vitro. The reasons for its discrepancy come from the different excipient compositions and formulation processes among different enterprises. In the process of drug development and production, the dissolution method can assist in the quality control of drugs. In this in vitro dissolution study, the dissolution profile determination method was established to investigate the dissolution behavior of the tablets produced by various enterprises in different dissolution media. Moreover, we sought to comprehensively evaluate the similarity of the dissolution profiles by comparison with the reference. The dissolution profiles of the seven generic formulations were similar to that of the reference. This evidence indicated the in vitro dissolution behavior of generic formulations was consistent with that of the reference. The dissolution behavior of TH tablets was not affected by excipient compositions and formulation processes.

### 3.2. In Vitro Permeability Experiment

Bioequivalence refers to the consistency of biological effects, mainly including the safety and effectiveness of clinical applications. It is the basic requirement for generic formulation research and development, as well as the replacement of clinical drug application [24]. The MacroFlux apparatus is a novel device that incorporates an absorption compartment into the dissolution apparatus. It has been shown that this device could be helpful in predicting the bioequivalence in vivo of drugs [14,22]. The µFlux apparatus is used to determine the permeability and intrinsic dissolution of a drug and explore the influence of excipients on drug permeability. Such data are crucial to the research and development process of new drugs and generic drugs.

In this study, the MacroFlux apparatus was used to determine the flux and total permeated amount of TH tablets from various enterprises. Moreover, it predicted the risk in the in vivo bioequivalence test. According to the in vivo fasting bioequivalence results of TH tablets from A and D enterprises (Table 6), the confidence interval of three in vivo parameters (C_max_, AUC_0–t_, and AUC_0–∞_) were within the range of 80–125%.

In the MacroFlux assay, the 90% confidence interval of flux and total permeated amount of TH tablets from Enterprises A and D ranged from 80–125% under FaSSIF conditions. This was consistent with the in vivo results. While satiety bioequivalence test of these two enterprises and the bioequivalence test of Enterprises B, C, E, and F were not conducted, the bioequivalence test of Enterprise G was conducted. The consistency of in vivo and in vitro results for Enterprises A and D indicated that the MacroFlux assay has a certain predictive ability for in vivo bioequivalence results.

The formulation processes and excipient compositions of TH tablets produced by each enterprise are shown in Table 7. According to Table 7, the TH tablets from Enterprise D were produced by direct powder compression, while the remaining TH tablets (Abbott Laboratories and Enterprises A, B, C, E, F, and G) were produced through wet granulation. The risk identified in the MacroFlux experiment may arise from the inconsistency in the processing method between various enterprises. There were certain differences between the compositions. To investigate whether the excipient compositions were responsible for the risk in the MacroFlux assay, the TH tablets for which risk existed in MacroFlux were ground into power to exclude the influence of different processing methods.

The MacroFlux experiment data of TH tablets were used to predict the bioequivalence, as well as to preliminarily determine the presence of risk in the bioequivalence test. The results of MacroFlux demonstrated that the TH tablets provided by Enterprises B, D, and G were likely to fail the bioequivalence test. Two aspects (API supplier and excipient compositions) were investigated to understand the causes of bioequivalence risk of TH tablets provided by these three enterprises. The results showed that the permeability of APIs was significantly lower for Enterprise I versus Enterprise II in pH 5.0 medium. Under other pH conditions, there was no significant difference in the permeability of APIs provided by Enterprises I and II. Among TH tablets provided by eight enterprises, the API of TH tablets produced by D was derived from I, while TH tablets produced by other companies were derived from II. Under the condition of pH 5.0, the permeability of TH API provided by Enterprise I was lower than that provided by Enterprise II. This evidence was consistent with the conclusion that TH tablets provided by Enterprise D were associated with a high risk in the MacroFlux FeSSIF experiment. It suggested that, in the FeSSIF experiment, the high risk of bioequivalence failure for TH tablets provided by Enterprise D may be due to the characteristics of the API.

We also sought to study the impact of excipient compositions. For this purpose, the formulations were ground into powder to avoid the influence of formulation processes, and the permeability of the drug powder was investigated. The results showed that there was no significant difference in the permeability of drug powder between different enterprises under the condition of FaSSIF and FeSSIF. These data suggested that the excipient compositions were not responsible for the bioequivalence risk of TH tablets provided by Enterprises B, D, and G. Regarding the TH tablets provided by Enterprises B and G, the API and formulation process are consistent with those of the reference formulation. The TH tablet produced by Enterprise B showed differences in the MacroFlux experiment and was shown to be equivalent to the reference formulation in beagle dogs. In summary, the present evidence shows that, under identical conditions (i.e., same API and formulation process), differences in the excipients of BCS Class I drugs will not alter the absorption of the preparation (i.e., the TH tablets provided by Enterprise B). Furthermore, differences in the API and formulation process may lead to differences in the absorption of the preparation (i.e., the TH tablets provided by Enterprise D).

The PAMPA method was suitable for investigating the passive diffusion in drug permeation. Therefore, there is a limitation of the PAMPA method. When API or excipients are not compatible with real-time UV measurements, another method, e.g., HPLC-UV/MS, needs to be used to determine the concentration in the donor chamber.

## 4. Materials and Methods

### 4.1. Instruments

The instruments used in this study included: µFlux and MacroFlux drug permeability apparatus (Pion Inc., Billerica, MA, USA); Agilent 708-DS dissolution apparatus and Agilent 1260 series high-performance liquid chromatography (HPLC) system (Agilent Technologies, Santa Clara, CA, USA); and Mettler XPEX206 1/100,000 electronic scales and Mettler Toledo pH meter (Mettler Toledo, Greifensee, Switzerland).

### 4.2. Agents

The structure of TH (459.92 g/mol) is shown in Figure 4. TH reference tablet (2 mg) was provided by Abbott Laboratories (Shanghai, China), and generic tablets (named a–g, 2 mg) were provided by each enterprise (A–G). TH standard and prazosin hydrochloride standard substances were purchased from the National Institutes for Food and Drug Control (Beijing, China). API (named i and ii) of TH were obtained from Enterprises I and II. Simulated intestinal fluid (SIF) powder was purchased from Beijing Niuniu Gene Technology Co., Ltd. (Beijing, China). Gastrointestinal tract (GIT) lipid solution and Prisma^HT^ buffer were obtained from Pion Inc. Buffer components (NaH_2_PO_4_, Na_2_HPO_4_, NaCl, NaOH, HCl, CH_3_COOH, CH_3_COONa, KH_2_PO_4_, KOH) and dimethyl sulfoxide were purchased from China National Medicines Corporation Ltd. (Beijing, China).

### 4.3. In Vitro Dissolution Study

#### 4.3.1. Experiment Conditions of Dissolution

The media used for the dissolution study (water, 0.1 N HCl, acetate buffer pH 4.5, phosphate buffer pH 6.8) were prepared according to the method described in the Guidelines for the Determination and Comparison of Dissolution Profiles of Ordinary Oral Solid Preparations [26]. The experiment was performed at 37 °C using 900 mL of dissolution medium. The rotational speed of the paddle was set at 50 rpm. After 5, 10, 15, 20, 30, 45, and 60 min, 5 mL of the solution were obtained and filtered through 0.45 µm pore size filters to prepare the test solution. The fresh corresponding volume of dissolution medium was replenished each time. Analysis was performed by HPLC, and all experiments were conducted six times.

#### 4.3.2. Chromatographic Conditions

Drug quantification in the dissolution samples was performed by HPLC-ultraviolet. Analytical HPLC procedures were modifications of those presented in Chinese Pharmacopoeia 2020 Vol II [27]. A reversed-phase ZORBAX Eclipse Plus (Agilent Technologies) C18 column (150 × 4.6 mm, 5 μm) was used. The mobile phase consisted of acetonitrile and perchloric acid (prepared using 2 mL of triethylamine in 1000 mL of water; the pH was adjusted to 2.0 with perchloric acid) 20:80 (volume/volume). The detection wavelength was 246 nm, and the injection volume was 20 µL. The flow rate was set at 1 mL/min. The elution time of TH was 7.5 min. Drug quantification was performed based on an external standard method. Two TH API standards (1 and 2) were prepared with corresponding medium (concentration: 2 μg/mL).

#### 4.3.3. Calculation of In Vitro Dissolution Data

The cumulative dissolution of TH at each sampling time point was calculated based on Equation (1). The dissolution profiles of different media were drawn according to the corresponding cumulative dissolution.
(1)$$\mathrm{Cumulative}\;\mathrm{dissolution}\;(\%)=\frac{A\times N\times f\times V_{m}+(A_{1}\times N_{1}+A_{2}\times N_{2}+\cdots +A_{t-1}\times N_{t-1})\times f\times V_{s}}{S}\times 100\%$$
(2)$$f=\left(\frac{C_{s1}}{A_{s1}}+\frac{C_{s2}}{A_{s2}}\right)÷2$$
where *A* is the peak area of TH in the test solution; *N* is the diluted multiples of the test solution, *f* is the correction factor of TH standard solutions; *Vm* and *Vs* are the volume of dissolution medium and sample, respectively; *t* is the number of sampling time points; *S* is the specification of TH (2 mg); and *Cs*1, *As*1, *Cs*2, and *As*2 are the concentration and peak area of TH API standard solutions 1 and 2, respectively.

### 4.4. MacroFlux Assay for Evaluating the Permeation of TH Tablets

The MacroFlux device was assembled as shown in Figure 5. It comprised the dissolution vessel (donor) and an acceptor chamber with membrane stirrer and fiber-optic probes. An artificial membrane (polyvinylidenfluoride: 0.45 μm, 3.69 cm^2^) impregnated with 50 μL GIT lipid solution was separating the donor from the acceptor containing 16 mL of pH 7.4 buffer. Donor and acceptor stirring were set at 50 and 250 rpm, respectively. At the beginning of the simulated fasting environment, pH 1.6 buffer (900 mL) simulating gastric conditions was added into the donor dissolution vessel; after 30 min, the medium in the donor dissolution vessel was transformed into pH 6.5 FaSSIF (pH 6.5) by adding the SIF solution. For the fed condition, pH 5.0 FeSSIF (900 mL) was used, which was added to the donor dissolution vessel without using a gastric stage.

Fiber-optic probes connected to a Pion Rainbow™ Dynamic Dissolution Monitor^®^ system were used to detect changes in UV spectra (200–720 nm), which were used to measure the drug concentration in the dissolution vessel and the acceptor chamber. The flux and the total permeated amount were calculated using Equations (3) and (4).
(3)$$J/\mathrm{Flux}=\frac{dc}{dt}\times \frac{V}{A}$$
(4)$$\mathrm{Total}\;\mathrm{permeated}\;\mathrm{amount}=C_{max}\times V$$
where *J*/*Flux* (µg·min^−1^·cm^−2^) is the total amount of material (or mass) crossing one unit area of the membrane per unit time; $\frac{dc}{dt}$ is the slope of the concentration-time profile in the acceptor chamber (µg· mL^−1^·min^−1^); *V* is the volume of the acceptor chamber (mL); *A* is the membrane area (cm^2^); and *C_max_* is the concentration of drugs in the acceptor chamber at the last time point (µg·mL^−1^).

### 4.5. µFlux Assay for Evaluating the Permeability of API

Using the µFlux device as shown in Figure 6, 16 mL of pH 5.0, 6.5, and 7.4 phosphate buffer solution and Prisma^HT^ buffer were accurately added into the donor chamber and the acceptor chamber, respectively. The two chambers were separated by a 1.65 cm^2^ membrane impregnated with 25 μL of GIT lipid solution to form a lipophilic barrier between the donor and acceptor chambers. The drug concentration in the donor and acceptor chambers was measured by fiber-optic probes, and the permeability of the two APIs of TH was calculated according to Equation (5) under the conditions of pH 5.0, 6.5, and 7.4.
(5)$$Pe_{\;}=Flux÷C_{t}÷60$$
where *P_e_* is the permeability of the drug (cm·s^−1^), and *C_t_* is the average drug concentration in the donor chamber (µg·mL^−1^).

### 4.6. µFlux Assay for Evaluating the Influence of Excipient

The excipient profile of TH tablets produced by eight enterprises are shown in Table 7.

To investigate whether the excipient prescriptions affected the flux and total permeated amount of TH tablets, formulations from different enterprises were ground into powder in this experiment. Of note, grinding eliminates the influence of the formulation process. The permeability was measured using a µFlux apparatus under FaSSIF and FeSSIF conditions. The reference formulation of TH tablet and generic formulations of TH tablets produced by Enterprises B, D, and G were respectively weighed. Subsequently, the permeability of these four batches of formulations after grinding into powder was measured by the µFlux apparatus.

Using the device as shown in Figure 6, the precisely weighed tablet powder was placed in the donor chamber. Using the method described in Section 2.5, the two chambers were isolated with the membrane with an area of 1.54 cm^2^. The media included in the donor and acceptor chambers were consistent with those used in the MacroFlux assay of tablets, while the volume was 20 mL.

### 4.7. Bioequivalence Studies in Dogs

#### 4.7.1. Experimental Method

Six beagle dogs (male and female: n = 3, respectively) were randomized following a double-cross design. The dogs received oral treatment with a single dose of TH tablets, including generic formulations and a reference formulation. Venous blood samples (2 mL) were collected at different time points (0, 0.25, 0.5, 1.0, 2.0, 3.0, 4.0, 6.0, 8.0, 12.0, 24.0, 36, and 48 h). Plasma was separated by centrifugation at 4000 rpm for 10 min. Thereafter, in each plasma sample (50 μL), prazosin hydrochloride (50 μL) (internal standard working solution; 50 ng·mL^−1^) and methanol precipitator (100 μL) were added. After mixing for 30 s and centrifugation at a high speed (14,000 rpm × 5 min) twice at 4 °C, supernatant (5 μL) was obtained for analysis.

#### 4.7.2. Chromatographic and Mass Spectrometry Conditions

A reversed-phase ZORBAX SB (Agilent Technologies) C18 column (100 × 2.1 mm, 3.5 μm) was used. The mobile phase was composed of water (containing 0.2% formic acid and 5 mmol·L^−1^ ammonium acetate) as Phase A and acetonitrile (containing 0.2% formic acid) as Phase B. The flow rate was set at 0.25 mL·min^−1^, and the column temperature was 40 °C. The gradient conditions and mass spectrum conditions of the liquid phase are shown in Table 8 and Table 9.

### 4.8. Statistical Analysis

#### 4.8.1. Comparison of In Vitro Dissolution Profiles

In the four dissolution media, if the cumulative dissolution of TH reference and generic formulation is not <85% at 15 min, the dissolution profiles can be considered similar. If the cumulative dissolution of the generic formulation is <85% at 15 min, comparisons of the dissolution profiles were performed using the f_2_ (similarity factor), according to Equation (6) [22]:(6)$$f_{2}\;=50\times log\left\{{\left[1+\left(\frac{1}{n}\right)\sum_{t=1}^{n}{\left(R_{t}-T_{t}\right)}^{2}\right]}^{-0.5}\times 100\right\}$$
where${\;R}_{t}$ and $T_{t}\;$ are the cumulative dissolution of the reference and generic formulations at time *t*, and n is the number of sampling points.

#### 4.8.2. PAMPA Experiments

In MacroFlux experiments, bioequivalence was predicted by calculating the 90% confidence interval for flux and total permeated amount. The *t*-test of IBM SPSS Statistics Version 22 software (IBM Corp., Armonk, NY, USA) was used for data analysis in API and excipient experiments. For bioequivalence studies in vivo, the 90% confidence interval of the area under the curve (AUC) or C_max_ was calculated, which was used to assess the bioequivalence of the generic drug in dogs.

## 5. Conclusions

The above experiments demonstrated the in vivo predictive power of the MacroFlux assay by comparing the in vitro flux to in vivo bioequivalence study results of the reference and generic formulations of TH tablets. In addition, there was a significant discrepancy in the permeability of TH APIs from various sources under the condition of pH 5.0. In the follow-up excipient experiments, it was manifested that different excipient compositions did not have a significant effect on drug permeability. Additionally, in the bioequivalence experiment in dogs, the TH tablets provided by Enterprise B showed equivalence to the reference. According to the above experimental results, differences in excipients do not lead to differences in the absorption of tablets when the sources of the API and formulation process were consistent with those of the reference formulation. Moreover, discrepancies in the API and formulation process may be the reason responsible for the difference in the absorption of tablets. The flux used in this experiment is only suitable for determining differences in the drug permeability of pharmaceutical formulations, APIs, and excipients; experimental exploration of formulation technology cannot be conducted. Therefore, further research on the formulation process is warranted.

## Figures and Tables

**Figure 1 pharmaceuticals-17-01024-f001:**
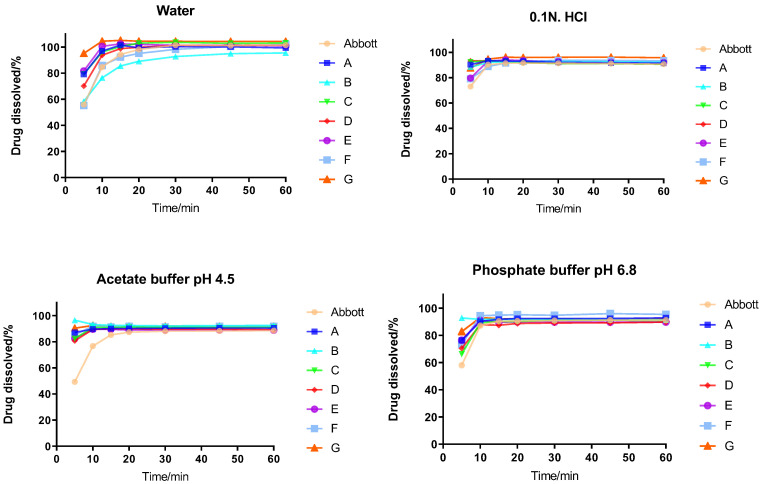
Dissolution profiles of each enterprise in four media.

**Figure 2 pharmaceuticals-17-01024-f002:**
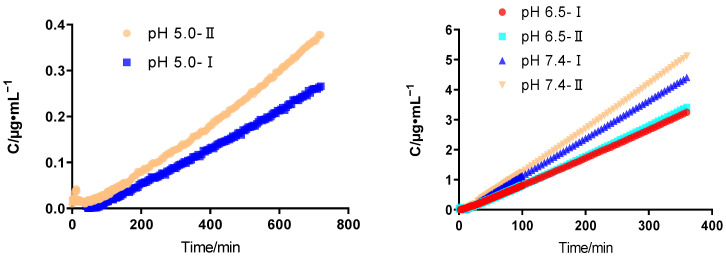
Permeation curves of TH API provided by Enterprises I and II under pH 5.0, 6.5, and 7.4. API: active pharmaceutical ingredient; TH: terazosin hydrochloride.

**Figure 3 pharmaceuticals-17-01024-f003:**
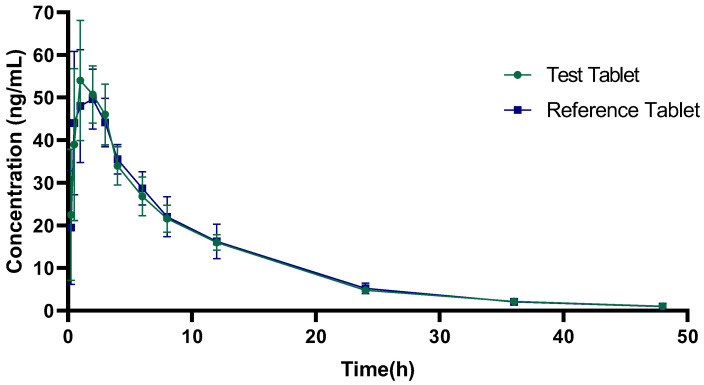
Blood concentration-time curve of six beagle dogs after oral administration of terazosin hydrochloride (TH) generic formulation and reference formulation.

**Figure 4 pharmaceuticals-17-01024-f004:**
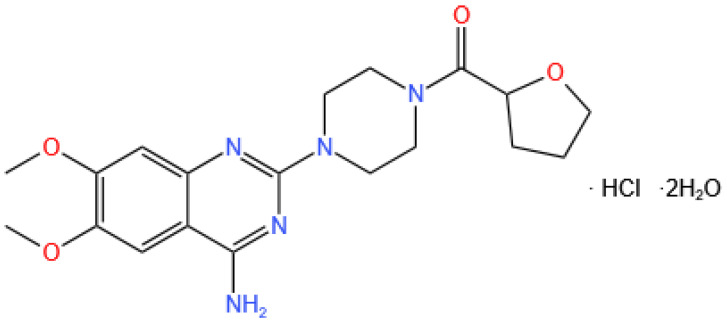
Structure of terazosin hydrochloride.

**Figure 5 pharmaceuticals-17-01024-f005:**
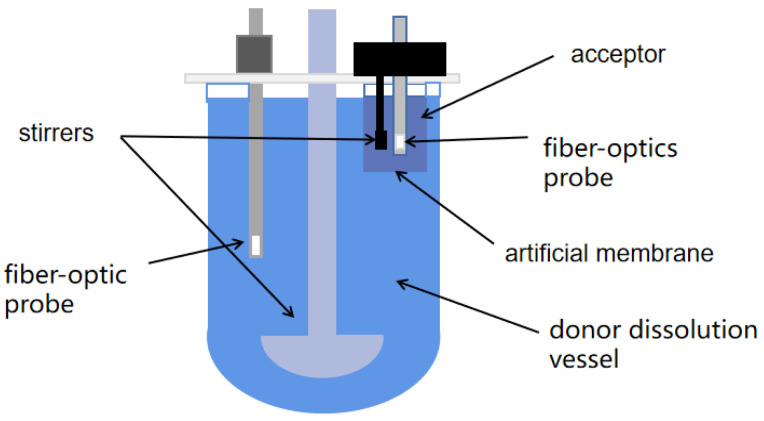
Schematic diagram of MacroFlux apparatus.

**Figure 6 pharmaceuticals-17-01024-f006:**
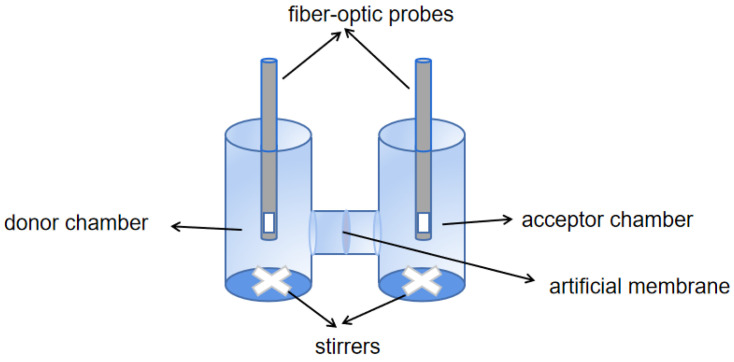
Schematic diagram of a μFlux apparatus.

**Table 1 pharmaceuticals-17-01024-t001:** MacroFlux test results of TH tablets.

Enterprise	FaSSIF		FeSSIF	
	Flux (µg·min^−1^·cm^−2^)	Total Permeated Amount (µg)	Flux (µg·min^−1^·cm^−2^)	Total Permeated Amount (µg)
Abbott Laboratories	0.00801 ± 0.00016	5.82 ± 0.03	0.00165 ± 0.00018	1.31 ± 0.21
A	0.00729 ± 0.00018	5.32 ± 0.23	0.00153 ± 0.00007	1.32 ± 0.06
B	0.00696 ± 0.00033	5.20 ± 0.20	0.00148 ± 0.00009	1.07 ± 0.08
C	0.00752 ± 0.00030	5.36 ± 0.15	0.00166 ± 0.00010	1.29 ± 0.09
D	0.00782 ± 0.00033	5.56 ± 0.16	0.00166 ± 0.00020	1.39 ± 0.13
E	0.00705 ± 0.00021	5.11 ± 0.22	0.00152 ± 0.00005	1.21 ± 0.04
F	0.00786 ± 0.00030	5.77 ± 0.36	0.00149 ± 0.00008	1.24 ± 0.06
G	0.00667 ± 0.00028	5.46 ± 0.47	0.00142 ± 0.00013	1.24 ± 0.17

Data are presented as the mean ± standard deviation (n = 3). FaSSIF: fasted state simulated intestinal fluid; FeSSIF: fed state simulated intestinal fluid; TH: terazosin hydrochloride.

**Table 2 pharmaceuticals-17-01024-t002:** The 90% confidence intervals for the flux and total permeated amount of TH tablets produced by each enterprise (n = 3).

Enterprise	FaSSIF		FeSSIF	
	Flux (µg·min^−1^·cm^−2^)	Total Permeated Amount (µg)	Flux (µg·min^−1^·cm^−2^)	Total Permeated Amount (µg)
A	87.20–94.60%	84.81–98.10%	85.22–99.70%	93.07–108.56%
B	79.83–93.76%	83.46–95.14%	80.07–98.96%	71.25–92.24%
C	87.48–100.23%	87.83–96.45%	90.86–110.33%	86.33–110.85%
D	90.61–104.56%	91.00–100.07%	79.53–121.15%	89.22–123.69%
E	83.59–92.26%	81.47–94.16%	86.74–97.39%	87.23–98.13%
F	91.82–104.37%	88.81–109.42%	82.12–98.17%	87.40–103.00%
G	77.39–89.14%	80.26–107.57%	72.42–99.78%	73.30–116.15%

FaSSIF: fasted state simulated intestinal fluid; FeSSIF: fed state simulated intestinal fluid; TH: terazosin hydrochloride.

**Table 3 pharmaceuticals-17-01024-t003:** P_e_ of TH API under pH 5.0, 6.5, and 7.4.

pH	Enterprise	P_e_ (10^−6^ cm·s^−1^)	*p*-Value (*t*-Test)
5.0	I	2.61 ± 0.264	0.032 *
	II	3.41 ± 0.333	
6.5	I	75.67 ± 2.487	0.234
	II	78.70 ± 2.807	
7.4	I	82.54 ± 5.939	0.091
	II	100.69 ± 12.885	

* *p* < 0.05 indicates significant differences in the P_e_ of API between the two enterprises. Data are presented as the mean ± standard deviation (n = 3). API: active pharmaceutical ingredient; P_e_: permeability; TH: terazosin hydrochloride.

**Table 4 pharmaceuticals-17-01024-t004:** P_e_ of TH tablet powder from eight enterprises under FaSSIF and FeSSIF conditions (n = 3).

Condition	Enterprise	P_e_ (10^−6^ cm·s^−1^)	*p*-Value (*t*-Test)
FaSSIF	Abbott Laboratories	35.42 ± 3.02	-
	B	33.85 ± 2.16	0.503
	G	31.27 ± 1.99	0.117
FeSSIF	Abbott Laboratories	4.74 ± 0.32	-
	B	5.14 ± 0.39	0.239
	D	5.20 ± 0.38	0.186
	G	5.70 ± 0.72	0.104

FaSSIF: fasted state simulated intestinal fluid; FeSSIF: fed state simulated intestinal fluid; Pe: permeability; TH: terazosin hydrochloride.

**Table 5 pharmaceuticals-17-01024-t005:** Main pharmacokinetic parameters of TH tablets in beagle dogs.

Enterprise	AUC_(0–t)_ (ng·mL^−1^·h^−2^)	AUC_(0–∞)_ (ng·mL^−1^·h^−2^)	C_max_ (ng·mL^−1^)	T_max_ (h)	t_1/2β_ (h)
Abbott Laboratories	545.27 ± 70.76	558.18 ± 72.66	55.68 ± 10.80	1.58 ± 0.66	8.18 ± 0.86
B	535.50 ± 77.18	554.83 ± 76.97	56.49 ± 9.65	1.17 ± 0.41	9.14 ± 1.76

AUC: area under the curve; C_max_: maximum drug concentration; TH: terazosin hydrochloride; T_max_: time to reach maximum concentration following drug administration; t_1/2β_: terminal elimination half-life.

**Table 6 pharmaceuticals-17-01024-t006:** Fasting bioequivalence study results for Enterprises A and D [25].

Enterprise	Parameter	T	R	T/R (%)	90% Confidence Interval
A	C_max_	49.36	50.04	97.68	91.05~104.79%
	AUC_0–t_	481.3771	484.7605	99.30	96.44~102.25%
	AUC_0–∞_	493.9153	496.4887	99.48	96.66~102.39%
D	C_max_	47.225	43.812	107.79	98.39~118.09%
	AUC_0–t_	411.6	406.7	101.23	96.17~106.55%
	AUC_0–∞_	424.7	419.9	101.14	96.16~106.38%

AUC: area under the curve; C_max_: maximum drug concentration.

**Table 7 pharmaceuticals-17-01024-t007:** Excipients and formulation processes of TH tablets from eight enterprises.

Enterprise	Composition of the Tablets	Formulation
Abbott	Lactose, corn starch, magnesium stearate, talcum powder	Wet granulation
A	Lactose, ethanol, magnesium stearate, MCC, pre-gelatinized starch	Wet granulation
B	Lactose, magnesium stearate, MCC, CMS-Na	Wet granulation
C	Lactose, corn starch, 10% corn starch pulp, magnesium stearate	Wet granulation
D	Lactose, magnesium stearate, MCC, CMC, pre-gelatinized starch	Direct powder compression
E	Lactose, magnesium stearate, L-HPC	Wet granulation
F	Lactose, magnesium stearate, MCC, CMS-Na	Wet granulation
G	Lactose, corn starch, magnesium stearate, MCC, PVP-K30	Wet granulation

CMC: sodium carboxymethylcellulose; CMS-Na: sodium carboxymethyl starch; HPMC: hydroxypropyl methylcellulose; L-HPC: low-substituted hydroxypropyl cellulose; MCC: microcrystalline cellulose; PVP-K30: povidon-K30.

**Table 8 pharmaceuticals-17-01024-t008:** Gradient elution conditions for liquid chromatography.

Time (min)	Phase A (%)	Phase B (%)
0	80	20
1	80	20
1	10	90
4	10	90
4	80	20
5	80	20

**Table 9 pharmaceuticals-17-01024-t009:** Conditions and related parameters of mass spectrum.

Parameter	TH	Prazosin Hydrochloride
Scanning mode	Positive	Positive
Ion source	ESI	ESI
Spray voltage (V)	3500	3500
Shealth gas (psi)	35	35
Aux gas (psi)	10	10
Capillary temperature (℃)	400	400
Detection mode	SRM	SRM
Precursor ion (*m*/*z*)	388.083	384.027
Product ion (*m*/*z*)	290.090	231.040
Tube lens (V)	94	97
Collision energy (V)	25	40

ESI: electrospray ionization; SRM: elected reaction monitoring; TH: terazosin hydrochloride.

## Data Availability

The data presented in this study are available in this article.

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
