# Peer review of "Bioequivalence Analysis of Terazosin Hydrochloride Tablets Based on Parallel Artificial Membrane Permeability Analysis"

_pharmaceuticals, 2024, doi:10.3390/ph17081024_

Round 1

Reviewer 1 Report

Comments and Suggestions for Authors

In this study, the authors used the parallel artificial membrane permeability analysis (PAMPA) to study the bioequivalence of Terazosin Hydrochloride Tablets. The topic of the manuscript is interesting. However, I have some remarks and recommendations as follows:

1. Has the LC-MS method been previously optimised and validated? Please add relevant information.

2. Any study limitations should be presented and clearly explained.

3. There are some typos in the text. The text of the manuscript should be carefully checked.

Author Response

Q1:Has the LC-MS method been previously optimised and validated? Please add relevant information.

A1: Thank you very much for your detailed and precise advice. The LC-MS method was developed and validated in the bioequivalence study.of Terazosin tablets. The method had high specificity with no interference to the analysis in dog plasma. Good linearity was observed within the range of 1–200 ng/mL. Both inter- and intra-batch precision coefficient of variation (CV %) were below 15%, and the accuracy deviation was within ±10%. The matrix factor for terazosin determination was in the limits of 0.82~1.07, indicating no obvious impact of plasma matrix on the analysis. The analytes remained stable throughout the entire experimental period. The developed LC-MS method was suitable for the quantification of terazosin in the bioequivalence study of Terazosin Hydrochloride Tablets in dogs.

Q2. Any study limitations should be presented and clearly explained.

A2:In this study, the PAMPA was established based on the principle of concentration difference between the two sides of the donor chamber and the acceptor chamber.Therefore, the PAMPA method was suitable for investigating the passive diffusion in drug permeation. When API or excipients are not compatible with real-time UV measurements, another method, e.g., HPLC-UV/MS, needs to be used to determine the concentration in the donor chamber. These were added in the revised draft (Line 260-263, Page 9).

Q3. There are some typos in the text. The text of the manuscript should be carefully checked.

A3:Thank you for your kind reminder. The text of the manuscript has been carefully checked again. All grammatical and typographical errors have been corrected.

Reviewer 2 Report

Comments and Suggestions for Authors

The manuscript describes bioequivalence analysis of terazosin hydrochloride tablets based on parallel artificial membrane permeability analysis. The topic is relevant to the aim and scope of the Pharmaceuticals. The manuscript is well written and easy to follow. Some clarifications in the texts are needed. Overall, this manuscript meets the standard for acceptance after addressing the below comments:

1)      Why were the drug dissolved % measured only at pH 4.5 and 6.8, not including physiological pH?

2)      The meaning between Figure 4 and Table 6 is controversial. The drug dissolved % was higher at pH 4.5 than at pH 6.8, but permeability was lower at pH 5 than at pH 6.5. This phenomenon is contrary to the physical meaning that more dissolution generally leads to higher permeability. Why did the phenomenon happen?

3)      The results in Table 8 seem to be insufficient to understand. In general, at the condition with lower AUC, higher Cmax, and shorter Tmax, the terminal elimination half-life becomes shorter. The results of Table 8 are exactly opposite this generality. Please explain the reason why the opposite results were acquired.

4)      Figure 5 is little visible. Specifically, the axis captions and the figure legends are too small.

Author Response

Q1:Why were the drug dissolved % measured only at pH 4.5 and 6.8, not including physiological pH?

A1: According to the Guidelines for the Determination and Comparison of Dissolution Profiles of Ordinary Oral Solid Preparations (https://www.nmpa.gov.cn/xxgk/ggtg/ypggtg/ypqtggtg/20160318210001725.html), Water, 0.1N HCl, pH4.5 acetate buffer and pH6.8 phosphate buffer were selected as dissolution media. And TH tablets were mainly dissolved in the stomach, so the pH of the dissolution medium chosen was acidic.

Q2: The meaning between Figure 4 and Table 6 is controversial. The drug dissolved % was higher at pH 4.5 than at pH 6.8, but permeability was lower at pH 5 than at pH 6.5. This phenomenon is contrary to the physical meaning that more dissolution generally leads to higher permeability. Why did the phenomenon happen?

A2: The drug dissolved % of TH Tablets at pH 4.5 and pH 6.8 conditions are both >85% at 15 min, and >90% at 60 min. There was no significant difference in dissolution rate between the two pH conditions. Terazosin is a weakly alkaline drug with the value of pKa is 7.05. Under weakly alkaline conditions, the API of terazosin is less dissociative and easier to absorb. So the permeability of terazosin in the pH 6.5 medium was higher than that in the pH5.0.

Q3: The results in Table 8 seem to be insufficient to understand. In general, at the condition with lower AUC, higher Cmax, and shorter Tmax, the terminal elimination half-life becomes shorter. The results of Table 8 are exactly opposite this generality. Please explain the reason why the opposite results were acquired.

A3: Terazosin is a highly soluble and osmolar drug with a bioavailability of about 90% in vivo. In this experiment, the reason for the low AUC exposure in animals was the low dose of the drug, 2mg/1 tablet/animal, which was about 0.2mg/kg according to the weight of the animal. After drug administration, the blood concentration in the body was low, and the maximum Cmax was only 40-70ng/mL. Therefore, this drug did not belong to the low exposure AUC, high Cmax drugs. The half-life t1/2 measured in this study is 8.18-9.14h, which was also consistent with the results reported in previous studies. (Li X Y; Chen X Y; Gu Q; et al. Determination of terazosin in Beagle plasma by LC/MS/MS and its application in pharmacokinetics study. Journal of Pharmaceutical Analysis.2004, 1, 14-17)

Q4: Figure 5 is little visible. Specifically, the axis captions and the figure legends are too small.

A4: Thank you for your kind suggestion. The Figure 5 (Now is Figure 2) has been modified.

Reviewer 3 Report

Comments and Suggestions for Authors

The manuscript is very well written but some minor issues should be improved.

2.7. Bioequivalence Studies in Dogs 190 2.7.1. Experimental method: The author should provide the ethical approval number for the animal experiments.

Figure 5: The quality of Figure 5 should be improved. The axis titles are not visible. 

Comments on the Quality of English Language

The manuscript should be thoroughly checked for grammatical errors.

Author Response

Q1: 2.7. Bioequivalence Studies in Dogs 190 2.7.1. Experimental method: The author should provide the ethical approval number for the animal experiments.

A1: We appreciate your suggestion. And the ethical approval number for the animal experiments is 00008523. It has been provided in the Institutional Review Board Statement.

Q2: Figure 5: The quality of Figure 5 should be improved. The axis titles are not visible. 

A2: Thank you for your kind suggestion. The Figure 5 (Now is Figure 2) has been modified.

Reviewer 4 Report

Comments and Suggestions for Authors

Dear Authors, please respond to:

1. Are you sure that terazosin is a BCS class 1?

2. Why did you use the µFlux apparatus only for the non-bioequivalent formulations?

3. For both the MacroFlux and µFlux setups: How do you know that the GIT lipids were not leached from the artificial membrane?

4. How did you evaluate the membrane integrity? (Both setups)

5. Would an immediate-release formulation of terazosin not qualify for a biowaiver? It is implied from the dissolution profile analysis. The blood concentration-time profiles found in dogs were also equivalent (Figure 6)

6. Grinding of the tablets does not abolish the effects of formulation. You have not provided any physicochemical proof of this. What about thermal analysis, particle size distribution etc? You can induce new physicochemical properties by grinding. 

7. The composition of the formulation from Enterprise B was not provided. How can you state that grinding removes the effect of formulation?

8. Is it fair to compare the MacroFlux and µFlux instruments with one another – both require different pathlengths of diffusion and various degrees of turbulence? What about the saturation concentrations in both systems?

9.  Why was the drug concentration monitored over 250–720 nm? Provide a spectrum to show the applicability of the wavelength range.

10. Quality of Figure 5 is poor. Font size is too small.

11. What was the %assay of the API from Enterprise I and II?

12. What were the differences in the properties of terazosin from Enterprise I and II? (line 289)

13. What conclusions were drawn from the µFlux experiments? Section 5. Conclusions do not mention µFlux

Comments on the Quality of English Language

The English is of acceptable quality. Only minor editing is required i.e., check spelling and grammar once more.

Author Response

Q1: Are you sure that terazosin is a BCS class 1?

A1:According to data released jointly by NICHD and FDA (http://zy.yaozh.com/orangebook/NICHD_BCS.pdf), terazosin belongs to BCS class 1/3. And In the literature published by Leslie Z. Benet et al. (BDDCS Applied to Over 900 Drugs,AAPS J. 2011 Dec; 13(4): 519–547), terazosin belongs to the BDDCS class 1. In the alkaline environment of the human intestine, terazosin, as a weakly alkaline drug, had good permeability in the intestine.

Q2: Why did you use the µFlux apparatus only for the non-bioequivalent formulations?

A2: In this study, the MacroFlux apparatus was first used to carry out the test, and the experimental results preliminarily showed that terazosin hydrochloride tablets provided by B, D and G enterprises had the non-bioequivalence risk. In order to explore the cause of non-bioequivalence, µFlux apparatus was used to study API and the excipients prescription. So the µFlux only used for the non-bioequivalent formulations.

Q3: For both the MacroFlux and µFlux setups: How do you know that the GIT lipids were not leached from the artificial membrane?

A3: The equipment and consumables for examining the permeability of drugs were all commercially produced products, including GIT lipid. The main function of GIT lipid solution was to activate the artificial membrane. It was used to simulate an immobile intestinal barrier that was not leach from the surface of the artificial membrane.

Q4: How did you evaluate the membrane integrity? (Both setups)

A4: The artificial membrane was a commercially produced product, and the integrity test had been carried out on the same batch of products before leaving the factory. During the experiment, we added solution to the acceptor chamber and observed whether the solution seeped out to determine the integrity of the artificial membrane in MacroFlux and µFlux systems. (The acceptor chamber was connected to the artificial membrane)

Any significant leaching of the membrane would eventually lead to the loss of integrity of the membrane, which has not happened in this study. In the µFlux assay, the fluxes at different pH donor media follow the pH-partitioning hypothesis well, showing that the permeability was pH dependent, showing the integrity of the membrane. Also, the constant flux shows that the membrane maintained the pH difference between the donor and acceptor side, proving again that the membrane did not lose integrity. For MacroFlux, the lack of flux during the SGF phase of the assay and then the API permeation during the FaSSIF phase shows a behavior that aligns well with the pH-partitioning hypothesis and the ability of the membrane to preserve the pH difference between the donor and the acceptor side as well.

Q5: Would an immediate-release formulation of terazosin not qualify for a biowaiver? It is implied from the dissolution profile analysis. The blood concentration-time profiles found in dogs were also equivalent (Figure 6).

A5: The MacroFlux apparatus was used to predict the bioequivalence of formulation in the human body. In this study, the result of MacroFlux showed that some TH tablets may have the risk of inequivalence, suggesting that terazosin may have the risk of bioequivalence in human bioequivalence experiment.

Q6: Grinding of the tablets does not abolish the effects of formulation. You have not provided any physicochemical proof of this. What about thermal analysis, particle size distribution etc? You can induce new physicochemical properties by grinding. 

A6: The effect of formulation process can be eliminated by grinding (line 149). The µFlux assay was conducted on the powder obtained by grinding, and the main purpose was to investigate the consistency of the generic preparation and the reference preparation. The behavior in µFLUX was consistent with the MacroFlux behavior which assumes that there was no change in the sample characteristics.

Q7: The composition of the formulation from Enterprise B was not provided. How can you state that grinding removes the effect of formulation?

A7: In the reviese draft, the formulation from Enterprise B was provided. In fact, we can remove the effect of formulation process by grinding, not the effect of formulation. (Line 149)

Q8: Is it fair to compare the MacroFlux and µFlux instruments with one another both require different pathlengths of diffusion and various degrees of turbulence? What about the saturation concentrations in both systems?

A8: In the MacroFlux instrument, the donor chamber simulated the dissolution process of formulation in the body, and the acceptor chamber simulated absorption into the blood circulation. It studied the bioequivalence of formulations from a macro perspective. As for µFlux instrument, it explored the causes of non-bioequivalent from a microscopic perspective. The research purposes of the two instruments were not the same, so there were slight differences in the settings. However, due to the different volume of the donor chamber of the two systems, the saturation concentrations were various. But the saturation concentration had no connection with the experimental results of the two systems due to different research purposes, and the concentration used in the donor chamber in this study was not the saturation concentration.

Q9:  Why was the drug concentration monitored over 250720 nm? Provide a spectrum to show the applicability of the wavelength range.

A9: The drug concentration was not monitored over the range of 250-720 nm. The UV-Vis absorption was monitored in this range so the analyst has the flexibility to select the appropriate spectrum range after completing the assay to determine the accurate API concentration. It is corrected in the revised manuscript. (Line 326 and Line 344)

Q10: Quality of Figure 5 is poor. Font size is too small.

A10: We appreciate your suggestion and the Figure 5 (Now is Figure 2) has been modified.

Q11: What was the %assay of the API from Enterprise I and II?

A11: According to the instruction, the %assay of the API provided by enterprises I and II were 99.8%.

Q12: What were the differences in the properties of terazosin from Enterprise I and II? (line 289)

A12: According to the API synthesis route provided by the two enterprises, the solvents used in the refining process of terazosin API produced by enterprise I were ethanol, 18% dilute hydrochloric acid and purified water, and the Enterprise II were ethanol and hydrochloric acid. The different refining methods used by the two enterprises would lead to differences in particle size and shape of API, which would may lead to differences in the absorption behavior of formulations.

Q13: What conclusions were drawn from the µFlux experiments? Section 5. Conclusions do not mention µFlux.

A13: In this study, we explored the causes of bioequivalence risk from two aspects of composition of API permeability and excipient prescriptions. And the results showed that different sources of APIs may be the reason responsible for the difference in the absorption of tablets. It was mentioned in the conclusion sections. (Line 413-415 and Line 421-422)

Round 2

Reviewer 2 Report

Comments and Suggestions for Authors

All issues have been addressed.